# Arc Detection of Photovoltaic DC Faults Based on Mathematical Morphology

Lei Song [1], Chunguang Lu [1], Chen Li [1], Yongjin Xu [1], Jiangming Zhang [1], Lin Liu [2], Wei Liu [2] and Xianbo Wang [3,*]

[1] Marketing Service Center of State Grid Zhejiang Electric Power Co., Ltd., Hangzhou 311152, China; songlei502@126.com (L.S.); zjdl_lcg@163.com (C.L.); shinelee520@163.com (C.L.); xvyongj@126.com (Y.X.); zjming@zj.sgcc.com.cn (J.Z.)

[2] State Grid Hangzhou Xiaoshan District Power Supply Company, Hangzhou 311200, China; 13758159382@163.com (L.L.); lwei0714@163.com (W.L.)

[3] Hainan Institute of Zhejiang University, Sanya 572025, China

[*] Correspondence: xianbowang@zju.edu.cn

**Abstract:** With the rapid growth of the photovoltaic industry, fire incidents in photovoltaic systems are becoming increasingly concerning as they pose a serious threat to their normal operation. Research findings indicate that direct current (DC) fault arcs are the primary cause of these fires. DC arcs are characterized by high temperature, intense heat, and short duration, and they lack zero crossing or periodicity features. Detecting DC fault arcs in intricate photovoltaic systems is challenging. Hence, researching DC fault arcs in photovoltaic systems is of crucial significance. This paper discusses the application of mathematical morphology for detecting DC fault arcs. The system utilizes a multi-stage mathematical morphology filter, and experimental results have shown its effective extraction of fault arc features. Subsequently, we propose a method for detecting DC fault arcs in photovoltaic systems using a cyclic neural network, which is well-suited for time series processing tasks. By combining multiple features extracted from experiments, we trained the neural network and achieved high accuracy. This experiment demonstrates that our recurrent neural network (RNN) based scheme for DC fault arc recognition has significant reference value and implications for future research. The ROC curve on the test set approaches 1 from the initial state, and the accuracy on the test set remains at 98.24%, indicating the strong robustness of the proposed model.

**Keywords:** DC fault arc; feature extraction; mathematical morphology; recurrent neural network

## 1. Introduction

Photovoltaic (PV) systems have gained significant popularity as a renewable energy source due to their environmental benefits and potential for reducing reliance on fossil fuels. However, the issue of DC arc faults in PV systems has worsened due to equipment aging and external factors. Large-scale photovoltaic power plants can generate DC output voltages of several kilovolts. Gaps or spaces between cables or connecting devices under high voltage conditions lead to strong electric field emissions. This causes emitted electrons from the cathode surface to accelerate towards the anode, resulting in collisions and ionizations within the gap. Consequently, there is a sharp increase in charged particles and temperature, leading to the breakdown of the gap and the formation of an electrical arc [1]. The stable combustion of the arc generates temperatures exceeding thousands of degrees Celsius. Failure to promptly detect these arc faults may potentially result in severe fire incidents [2]. Statistical data shows that over 40% of fire accidents in photovoltaic power plants are caused by DC arcs. To address this issue, the National Electrical Code (NEC) in the United States, specifically in Article 690.11, requires photovoltaic systems with DC voltages exceeding 80V to be equipped with fault arc detection devices and circuit breakers [3].

Arc faults frequently occur in PV systems, and the sustained arc can generate high-temperature plasma that poses a significant risk of severe damage to system components [4–6]. PV DC arc damages can occur due to various reasons, including but not limited to faulty equipment, installation errors, or natural disasters. PV DC arc damages can occur due to various reasons, including but not limited to faulty equipment, installation errors, or natural disasters. In 2009, a fire incident in California, United States, caused damage to 1826 solar photovoltaic panels with a combined generating capacity of 383 kW [7]. The Netherlands reported 15 fire accidents associated with solar photovoltaic panels in 2009 [8]. Because hybrid energy systems, including PV power stations, are widely adopted in rural areas for independent power supply, and the PV DC arc can easily cause losses for residents [9]. In 2012, a warehouse in Goch, Germany, experienced a significant fire outbreak attributed to solar panels, resulting in an approximate burning area of 4000 square meters [10]. Therefore, the development of effective arc detection methods and standards is crucial for ensuring the safe and reliable operation of PV systems [11,12].

The photovoltaic DC detection method utilizes the characteristics of arc light, arc sound, and electromagnetic radiation to monitor fault arcs in photovoltaic systems [13–15]. This specialized approach employs dedicated sensors for detecting arc light, sound, and electromagnetic radiation generated by the arc. Photoelectric sensors detect arc light, sound sensors capture arc sound, and electromagnetic sensors receive electromagnetic radiation. By analyzing and processing these signals, fault arcs can be accurately detected and identified. Zhao et al. [16] proposed a method for detecting series arc faults in DC power systems. This method involves steady-state analysis in the frequency domain to identify series arc faults. Parameters such as the structure similarity index and 6 dB bandwidth box are calculated to extract the similarity of the steady-burning arc spectrum. This allows for the effective identification of arc faults and differentiation from normal operation. Xiong et al. [17] presented a method for detecting DC arc faults based on electromagnetic radiation signals, utilizing a designed DC arc generating device and a fourth-order Hilbert curve fractal antenna to analyze the amplitude and spectra of electromagnetic radiation signals. The test results demonstrate that the characteristic frequency of electromagnetic radiation signals can be utilized as a detection parameter for DC arc faults in PV systems, which have higher frequencies and longer pulse intervals compared to switch operations. In [18], a noninvasive arc fault detector based on magnetic-field sensing and autocorrelation algorithm is developed for DC microgrids. A multicharacteristics arc model is established based on the volt-ampere, current sag, and power spectral characteristics of arc faults. According to the frequency domain features of arc faults and interaction effects between different branches, the arc-detection-point selection principle is formed. Ke et al. [19] presented a novel method for detecting arc faults by leveraging the characteristics of electromagnetic radiation. Through the reception of electromagnetic radiation signals with comparable characteristic frequencies, this method enables accurate differentiation between operational arcs and fault arcs. It effectively mitigates the influence caused by non-linear loads and switch operations in the circuit, thus ensuring the prevention of false alarms and omissions. Li et al. [20] proposed a planar localization method that only requires two detection points to address the challenge of detecting and isolating arc faults in DC microgrids or photovoltaic systems. By forming a horizontal triangle with an antenna array and the fault source, DC arc faults can be effectively located. Signal pulses are distinguished and extracted using cross-correlation techniques, and a neural network along with the received signal strength indicator is employed to estimate the distance of the arc.

The method based on electromagnetic radiation primarily concentrates on investigating the radiation properties of arcs and conducting fault detection using spectral features. Nevertheless, this approach is heavily influenced by complex environmental factors and is also constrained by sensor limitations, which significantly restricts its effectiveness. The detection method based on the time-frequency domain characteristics of fault arc current and voltage is currently a more mainstream approach in direct current arc fault detection methods [21–24]. A large number of domestic and international scholars have conducted

extensive research in this field. Lu et al. [25] conducted an analysis of the variations in line current and power supply voltage resulting from DC fault arcs, considering the volt-ampere characteristics associated with these faults. They proposed a comprehensive method for detecting DC series fault arcs by utilizing information from the line current and power supply voltage. This approach involves examining the rate of decrease in detection current, average rate of current change, and standard deviation of the AC components present in the line current and power supply voltage, enabling the detection of fault arcs. In [26,27], a fault arc detection algorithm is developed by comparing the relative changes in current in the frequency spectrum and time series. Furthermore, studying the impedance of fault arcs through a small-signal model enables the determination of resonance frequencies in the low-frequency range under fault arc conditions. This impedance model can be utilized to design a frequency analysis range that effectively excludes inverter switching noise. Ahmadi et al. [28] applied the high-frequency component of normalized DC voltage to extract arc fault features, which effectively removes interference caused by inverter switch characteristics through lagged subtraction, and detects arc faults by comparing the power ratio (or signal-to-noise ratio) between the low-frequency component and the arc signal power. Chen et al. [29] presented a robust algorithm for identifying photovoltaic (PV) series arc faults amidst complex interferences, comprehensively understanding their features through various experiments, and using loop current signatures and quantificational evaluations to establish optimal detection variables. The algorithm achieves arc fault discovery through fusion coefficients and dynamically adjusts threshold values, demonstrating its effectiveness through experimental results on a simulated platform.

The drawbacks of arc fault detection methods based on time-frequency characteristics are shown in the following aspects: risks of false alarms and omissions, susceptibility to non-linear loads and switch operations, requiring additional hardware equipment, and parameter adjustments. In recent years, with the rapid advancement of pattern recognition, an increasing number of scholars have started to employ machine learning and deep learning techniques to assist in the detection of direct current arc faults [30–34]. Chen et al. [35] proposed a method using a multi-input CNN model with squeeze-and-excitation and inception networks to detect series arc faults in PV systems, achieving a high detection accuracy of 97.48%. The method effectively mitigates the influence of switching frequency, can detect faults in different locations, and withstands disturbances from dynamic shading, maximum power point tracking, and strong wind, providing a solution for rapid arc fault detection. Georgijevic et al. [36] introduced a quantum probability model-based arc-fault detection algorithm for PV systems that utilizes the modified Tsallis entropy of the PV panel current to differentiate between arc and no-arc states. The algorithm operates on a plug-and-play principle, requiring no prior knowledge of the PV system, and has been successfully tested in both simulated and real-world PV systems, demonstrating high sensitivity and robustness in detecting various types of series arcs without false detections. Qian et al. [37] introduced a practical adaptive method for detecting series DC arc faults in PV systems, utilizing the adjacent multi-segment spectral similarity (AMSSS) characteristic and principal component analysis (PCA) to establish an adaptive threshold model. The method is validated through tests conducted on a 20-module photovoltaic plant platform under various conditions, demonstrating strong arc detection capability and environmental adaptability, making it suitable for real-world PV systems. Wang et al. [38] explored the limitations of current-based series arc fault detection methods and introduced a new approach utilizing the reflection of fault information in the characteristic frequency band of arc voltage at the monitoring point, leading to the development of a comprehensive detection strategy based on voltage characteristic energy amplitude and phase mapping distribution distance for different load types. The experimental results demonstrate its effectiveness with varying line parameters and load types.

A method combining Gramian angular summation field (GASF) and squeeze and excitation-deep convolution generative adversarial network (SE-DCGAN) is proposed

for series arc fault (SAF) detection in PV arrays by accurately extracting transient current data, converting it into amplified GASF images, augmenting SAF samples, training a CNN for identification, and improving generalization through fusion training, achieving high recognition accuracy without misjudgments for interference events and demonstrating improved universality [39]. Et-taleby et al. [40] proposed a new model combining the convolutional neural network (CNN) for feature extraction and support vector machine (SVM) for classification to detect and classify faults in electroluminescence images of PV panels, achieving high classification performance.

Mathematical morphology is a mathematical analysis method that is based on the morphological changes of signals [41,42]. It is commonly used in various fields such as image processing, signal processing, and pattern recognition. This method involves defining structuring elements and applying morphological operations such as erosion, dilation, opening, and closing to extract features from signals and analyze them. Compared with the aforementioned method, the mathematical morphology-based photovoltaic DC arc fault detection method has higher accuracy and robustness. Its advantages lie in its ability to accurately extract fault features, suppress noise interference, and exhibit strong adaptability, reduced interference, and minimal training data requirements. Culjak et al. [43] proposed a fast fault detection method for radial DC microgrids, which is achieved through mathematical morphology denoising filters and local measurements. The method can withstand communication delays and faults, differentiate between different types of faults, and ensure the reliability of protective relays. It is cost-effective and applicable to digital signal processing hardware with real-time operating systems.

Compared to traditional methods, it offers advantages such as low sampling frequency, high fault tolerance, and robustness against noise. Gao et al. [44] investigated the time-frequency characteristics of photovoltaic arrays under normal and arc fault conditions, and proposed a novel diagnostic method for photovoltaic arc faults. The method utilizes the mathematical morphology-modified empirical wavelet transform algorithm to obtain the time-frequency domain matrix of the signal. Additionally, the fault features are characterized using composite multiscale permutation entropy, and binary classification is achieved through a twin support vector machine. The spectra obtained from twin support vector machine decomposition are smoothed using mathematical morphological closing operations to address the problem of densely packed frequency divisions in DC arc signal spectra. Gautam et al. [45] presented a method for detecting high impedance faults using mathematical morphology (MM) that can run in parallel with existing protection schemes. The proposed method is fast, reliable, and safe, and suitable for real-time applications. It utilizes voltage waveforms sampled at a substation for fault detection and has been successfully validated on different standard test feeders under various load and interference conditions. Furthermore, the low computational overhead inherent in MM-based tools provides an advantage for real-time applications. The detailed comparison of different operational strategies of photovoltaic DC detection is exhibited in Table 1.

This paper adopts a method based on multi-level mathematical morphology filters to investigate the application of mathematical morphology in the detection of direct current (DC) arc faults. This method can effectively extract the characteristics of fault arcs. In addition, we propose a PV system DC arc fault recognition scheme based on recurrent neural networks (RNNs). RNNs are suitable for handling tasks related to time series and are trained with multiple features extracted from experiments, achieving high accuracy in recognition.

The remainder of this paper is organized as follows. Section 2 outlines mathematical morphology theory. Section 3 illustrates the application of mathematical morphology on fault arc. Section 4 describes the experimental results for DC fault arc recognition based on deep learning. Section 5 concludes the contributions of this research and discusses possible future work.

**Table 1.** Literature reviews of operational strategies of photovoltaic DC detection.

| Detection Method | Key Techniques | Reference |
|---|---|---|
| Arc light, arc sound, and electromagnetic radiation | Similarity of the steady burning arc spectrum | [16] |
| | Fourth-order Hilbert curve fractal antenna | [17] |
| | Volt-ampere, current sag, and power spectral of arc faults | [18] |
| | Reception of electromagnetic radiation signals with comparable characteristic frequencies | [19] |
| | Planar localization method requiring two detection points | [20] |
| Time frequency domain characteristics of arc current and voltage | Examination of current decrease rate, current change average rate, and standard deviation of the AC line current and voltage | [25] |
| | Impedance of fault arcs through a small-signal model to determine resonance frequencies | [26] |
| | High-frequency component of normalized DC voltage to extract arc fault features | [28] |
| | Using loop current signatures and quantificational evaluations to establish optimal detection variables | [29] |
| Learning based pattern recognition algorithm | A multi-input CNN model with squeeze-and-excitation and inception networks | [35] |
| | A quantum probability model with Tsallis entropy | [36] |
| | Adaptive threshold model with AMSSS and PCA | [37] |
| | A comprehensive detection strategy based on voltage characteristic energy amplitude and phase mapping distribution distance | [38] |
| | GASF-GAN-CNN based transient current identification | [39] |
| | CNN-SVM based feature extraction and classification | [40] |
| Mathematical morphology | Mathematical morphology denoising filters and local measurements | [43] |
| | Mathematical morphology modified empirical wavelet transform algorithm | [44] |
| | Detecting high impedance faults using mathematical morphology (MM) | [45] |
| Ours | RNN-based mathematical morphology with higher accuracy in recognition | — |

## 2. Mathematical Morphology Theory

Mathematical Morphology (MM) was co-founded by French scientists Georges Matheron and Jean Serra in 1964, and it is based on set theory [46]. The fundamental idea of mathematical morphology is to use a "probe" called a structuring element to gather information from the signal being processed. By moving the probe through the signal, relationships between different parts of the signal can be examined, allowing for the extraction of useful global or local features. Mathematical morphology offers clear geometric interpretations, simple and fast operation processes, and ease of implementation in hardware, making it increasingly applied in the field of industrial information.

In the domain of fault arc studies, the application of mathematical morphology has been relatively limited. In this paper, the characteristics of mathematical morphology are combined with a brief introduction of its basic theory, aiming to explore its feasibility in the field of fault arcs.

### 2.1. Basic Operations of Mathematical Morphology

Mathematical morphology performs matching and modification operations on signal waveforms through the operations between the structural elements and the signal, such as erosion, dilation, opening, closing, etc. [47]. Mathematical morphology consists of two fundamental operations: erosion and dilation [26,28]. It is assumed that the original signal $f(n)$ and structural element $g(n)$ are discrete functions defined on sets $F = \{0, 1, \ldots, N-1\}$ and $G = \{0, 1, \ldots, M-1\}$, $N \geq M$, respectively.

The dilation and erosion operations of $f(n)$ with respect to $g(n)$ can be defined as follows:

$$(f \oplus g)(n) = \mathbf{max}\{f(n-m) + g(m)\} \tag{1}$$

where $m = \{0, 1, \ldots, M-1\}$.

$$(f \ominus g)(n) = \mathbf{max}\{f(n+m) - g(m)\} \tag{2}$$

where $\oplus$ and $\ominus$ represents expansion symbol and corrosion symbol, respectively.

Dilation and erosion are the fundamental operators in mathematical morphology, and their operations have an irreversible nature. Dilation and erosion operations can remove smaller details from the original signal based on the size of the structuring element while preserving the fundamental characteristics of the signal, resulting in a simplified structural representation of the signal. Erosion operation can be used to eliminate small and insignificant points, while dilation operation can be used to fill in gaps in waveforms.

*2.2. Mathematical Morphological Operators and Their Combinations*

The expansion and corrosion operations based on mathematical morphology can obtain other operators, and morphological algorithms with different characteristics and functions can be constructed based on basic operators [48,49]. Table 2 lists some widely used morphological operators and applications of multi-scalar morphology theory. $f(x)$ represents the signal to be processed and $g(x)$ represents the structuring element.

**Table 2.** Widely used mathematical morphology operators and methods.

| Name | Operational Formula | Remark |
|---|---|---|
| Open operation | $f \circ g = (f \ominus g) \oplus g$ | Filter the peak noise above the signal |
| Closed operation | $f \bullet g = (f \oplus g) \ominus g$ | Suppress the trough noise below the signal |
| Open-close operation | $O_c(f) = (f \circ g \bullet g)$ | The output amplitude is small |
| Closed-open operation | $O_c(f) = (f \bullet g \circ g)$ | The output range is too large |
| Top-hat operator | $T = f - f \circ g$ | Detection crest |
| Bottom-hat operator | $B = f - f \bullet g$ | Detection trough |
| Peak-valley probe operator | $D = 2f - f \circ g$ | Detect peak points and peak and valley points |
| Adaptive morphological filtering | $y(x) = a_1 O_c(f) + a_2 O_c(f)$ | Open-close and closed-open weighting coefficient adaptive and structural element adaptive |
| Morphological gradient calculation (MG) | $G_3 = (f \oplus g) - (f \ominus g)$ | Highlight the edge information |
| Multi-resolution Morphological Gradient Computing(MMG) | $\rho_{g^+}^a = \left( \rho_g^{a-1} \ominus g^+ \right) - \left( \rho_g^{a-1} \oplus g^+ \right)$ $\rho_{g^-}^a = \left( \rho_g^{a-1} \ominus g^- \right) - \left( \rho_g^{a-1} \oplus g^- \right)$ $\rho_g^a = \left( \rho_{g^+}^a + \rho_{g^-}^a \right)$ | More detailed transformations are made for rising and falling edges to show more subtle changes in the signal |
| Cascade Multi-resolution Morphological Gradient Computing (SMMG) | $F(x) \longrightarrow MF1 \longrightarrow MFn \longrightarrow SMMG(f)$ | The transient characteristics of the signal which are not obvious can be enhanced, and the generalized multiresolution gradient transform can be derived by increasing the width of structural elements |
| Multiscale morphology morphological spectrum | The multi-scale corrosion operation, expansion operation, open operation, and close operation are derived from the shape quantity distribution curve | Time domain transformation method based on multi-scale morphological analysis |

## 3. Application of Mathematical Morphology on Fault Arc

As a powerful tool for extracting information from signals, mathematical morphology can be used to detect small changes in the waveform of DC fault arc signals. The following content in this article will detect DC fault arc signals by constructing different morphological operators and conducting relevant result analysis.

### 3.1. Description of Experimental Environment and Equipment

### 3.1.1. Experimental Environment

As shown in Figure 1, this experiment site is a small photovoltaic grid-connected system built on the top floor of an enterprise in Wenzhou, Zhejiang Province, which is connected to the local power grid through a grid-connected inverter. The photovoltaic power generation system consists of 18 photovoltaic panels in series into one road, a total of three parallel into a bus box, and finally through the bus box into the inverter. The parameters of a single photovoltaic panel are the peak power of 260 W, the best voltage under the maximum power $U_m = 31.03$ V [50], the best working current $I_m = 8.38$ A, the open circuit voltage $U_{oc} = 38.66$ V, and the short circuit current $I_{sc} = 8.82$ A. The maximum output current of the photovoltaic array $I_{om} = 8.82 * 3 = 26.46$ A.

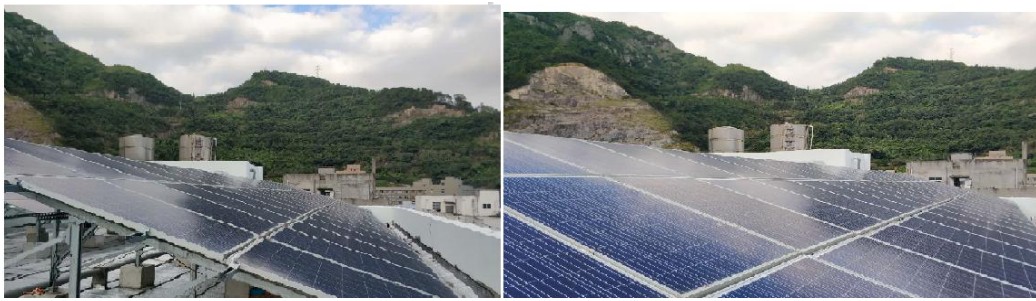

**Figure 1.** Photovoltaic array at the experimental site in Wenzhou, Zhejiang Province.

In this experiment, only one series circuit photovoltaic panel is selected as the power supply. Considering that a single photovoltaic panel is connected to other panels through connectors, the series fault arc mainly occurs between connectors, so this paper chooses to disassemble the connector between two photovoltaic panels and connect it to the arc-generating device for test.

### 3.1.2. Arc Generator

The arc generation device used in the experiment is designed with reference to the UL1699B standard. However, considering that in this standard is a sleeve between the two poles, we put a metal wire to help combustion, as shown in Figure 2. This device uses two electrodes of different shapes and materials, one is a cone and the other is a cylinder, and the material is a carbon rod and a metal rod. When the device is connected to the circuit and the current is turned on, the knob is adjusted so that the two electrodes, which were originally in contact, are slowly separated. When the electrodes are completely separated, an electric field is generated between the two poles due to the PV photovoltaic voltage, which breaks through the air to form an electric arc. The temperature rises between the arc gaps, thermal ionization continues to occur, the conductivity increases with it, the arc voltage decreases, and the arc burns steadily [19,25]. This method avoids randomness due to wire combustion and avoids blockage of the arc channel due to casing.

In the experiment, the current data is collected by the mangano-copper shunt connected in series in the photovoltaic circuit, the voltage at both ends of the shunt is measured, and the waveform data is converted by A/D after the signal conditioning circuit is transformed. This is shown in Figure 3.

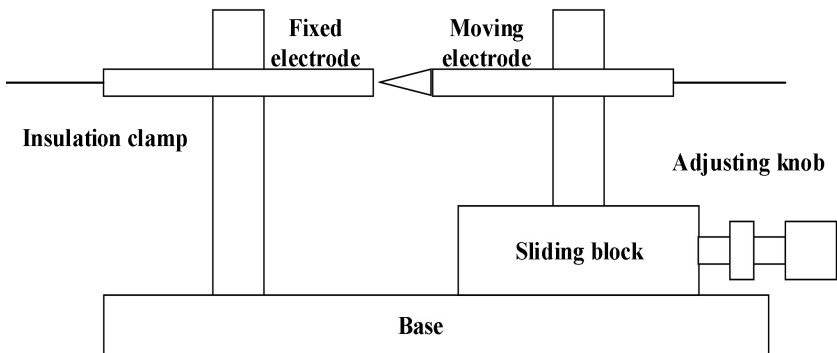

**Figure 2.** Arc generator used in the experiment.

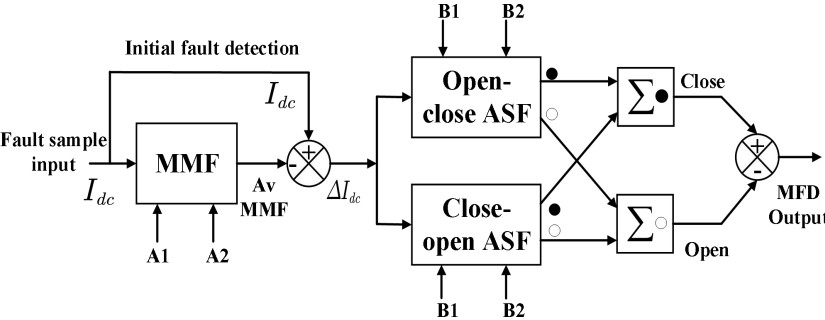

**Figure 3.** Multilevel mathematical morphology filter cascade composition.

### 3.2. Analysis of Fault Arc Using Multi-Level Mathematical Morphological Filters

By cascading mathematical morphological operators, signals can undergo various processing effects. In order to filter out noise in photovoltaic systems and extract the signal characteristics of fault arcs, this section designs a multi-level mathematical morphological filter, called Mathematical Morphological Fault Arc Detector (MFD), as shown in Figure 3 [29], through different combinations of mathematical morphological operators.

The structure consists of two main parts. The first part is composed of median filters in mathematical morphology (referred to as MMF), and the second part consists of alternating hybrid filters based on opening and closing operations (referred to as ASF). Four different structuring elements are used in this structure, denoted as $A_1, A_2, B_1, B_2$. Through experimental parameter adjustments, these four structuring elements are finally set as follows:

$$A_1 = B_1 = [0.995, 1, 0.995], \quad A_2 = B_2 = [0.957, 1, 0.957] \tag{3}$$

In this section, two types of current waveforms with current transients under normal conditions, represented as Figure 4a,c, were selected for MFD transformation, resulting in the obtained results shown in Figure 4b,d. To make a more intuitive comparison, a segment of the output signal after MFD transformation for a fault arc was selected and compared with the signals from Figures 4d and 5, as shown in Figure 6.

Comparing the output signals after the MFD transformation of the normal and fault arc current signals, as shown in Figure 4, it can be observed that the majority of the output signal after the MFD transformation for the normal condition is overshadowed by the output signal of the fault arc. The output values for the normal condition are mostly below 0.01, whereas for the fault arc condition, the outputs are generally between 0.01 and 0.05. Furthermore, most of the amplitudes are small, and the overall fluctuations are also relatively small. Statistical analysis can be utilized in the future to further differentiate between the two conditions.

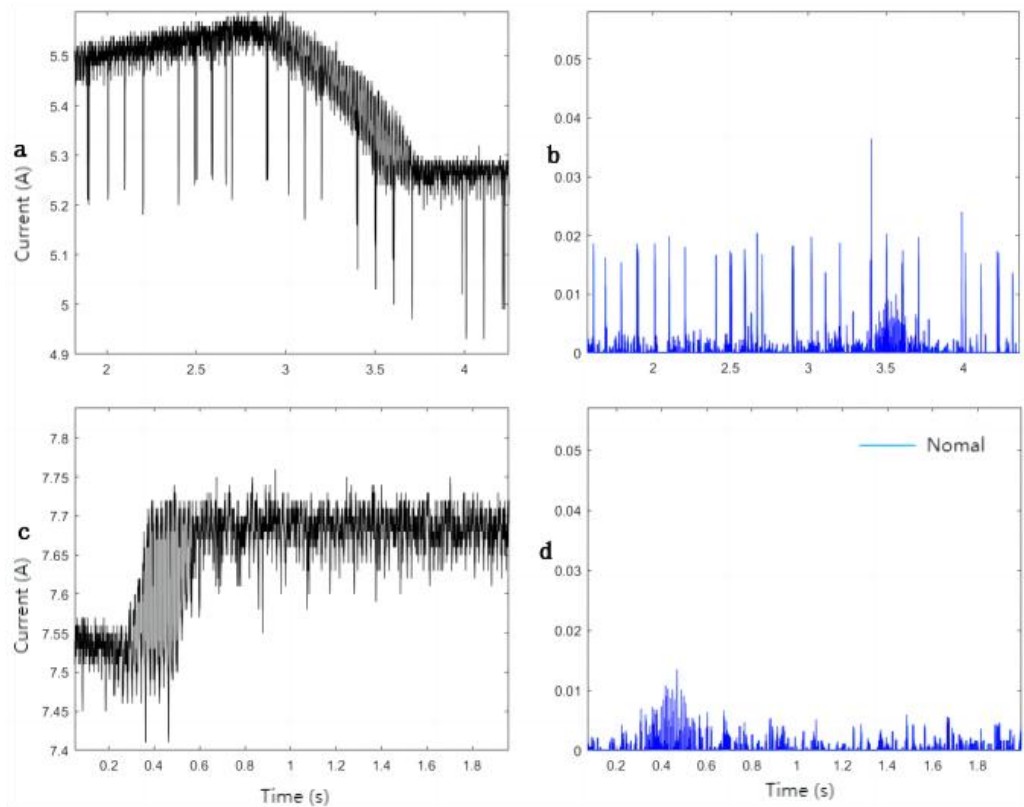

**Figure 4.** Output signals of the current waveform after MFD transformation: (**a**) Original fault arc current signal; (**b**) Output signal after MFD transformation of the current signal in (**a**); (**c**) Original normal current signal; (**d**) Output signal after MFD transformation of the current signal in (**c**).

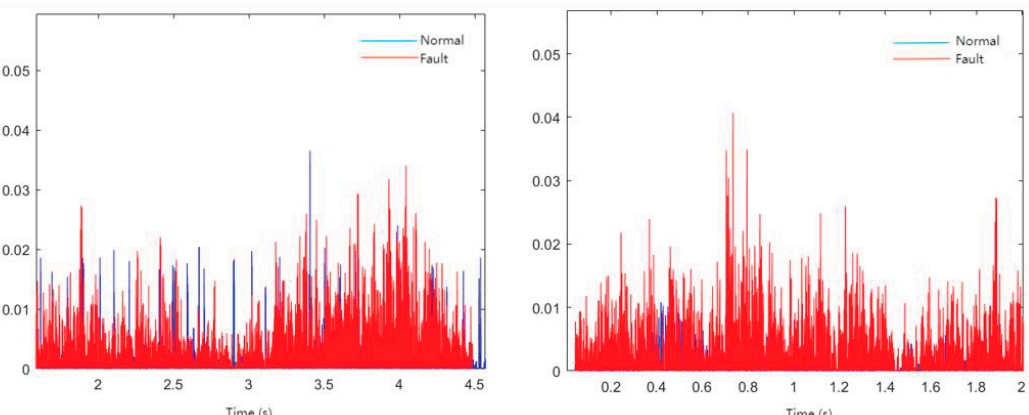

**Figure 5.** Comparison of the output signals after MFD transformation for the normal and fault arc current signals.

To perform a quantitative analysis of the output results, ten random samples were selected for both normal current and fault arc conditions. Using a period of 240 sampling points, the average amplitude and variance of the output signals were calculated. The results are shown in Tables 3 and 4. The output signals obtained after the operation of the multi-level mathematical morphological filter structure reflect the differences between fault arcs and normal currents when considering the average amplitude and variance. However, it should be noted that in practical photovoltaic systems, there may be more interference present. Therefore, the signals extracted by the multi-level mathematical morphological filters designed in this study cannot be solely relied upon to determine the presence of a

fault arc. However, these signals can serve as a characteristic feature of fault arcs, which can be used in conjunction with other features for fault detection and analysis.

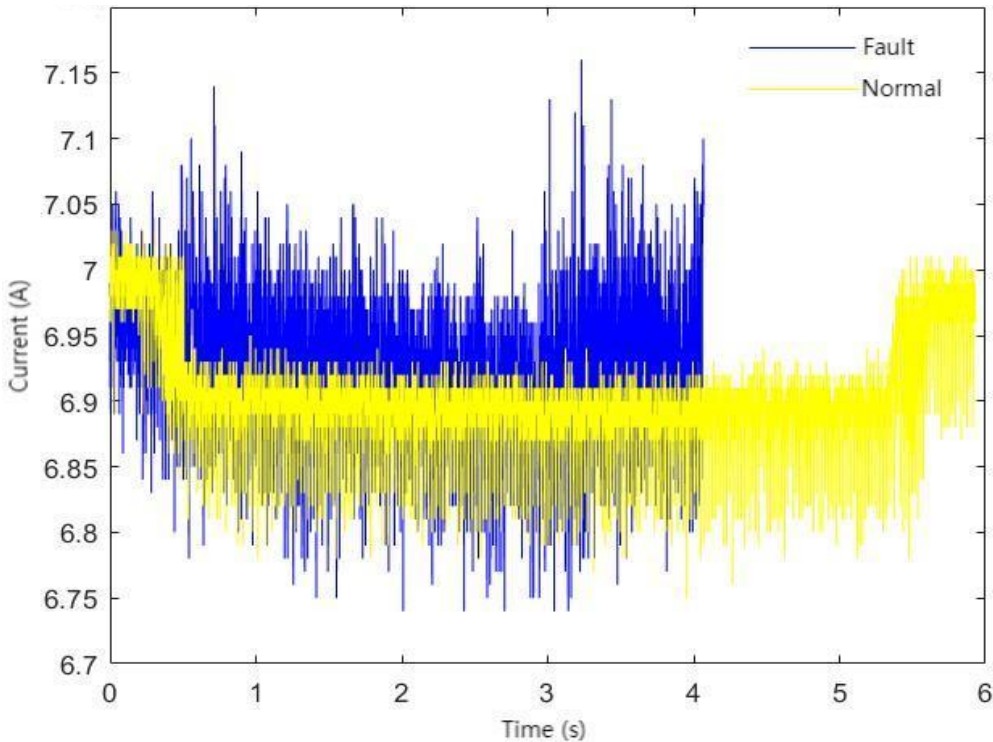

**Figure 6.** Waveforms of current under normal conditions and current when fault arcing occurs.

**Table 3.** Widely used mathematical morphology operators and methods.

| Group | 1 | 2 | 3 | 4 | 5 |
|---|---|---|---|---|---|
| Normal | 0.0013 | 0.0010 | 0.0015 | 0.0013 | 0.0014 |
| Arc Fault | **0.0086** | **0.0062** | **0.0072** | **0.0097** | **0.0066** |
| Group | 6 | 7 | 8 | 9 | 10 |
| Normal | 0.0014 | 0.0017 | 0.0032 | 0.0024 | 0.0035 |
| Arc Fault | **0.0072** | **0.0059** | **0.0098** | **0.0069** | **0.0078** |

**Table 4.** Widely used mathematical morphology operators and methods.

| Group | 1 | 2 | 3 | 4 | 5 | 6 | 7 | 8 | 9 | 10 |
|---|---|---|---|---|---|---|---|---|---|---|
| Normal $(10^{-6})$ | 4.08 | 9.06 | 5.27 | 1.75 | 2.04 | 2.78 | 8.01 | 1.39 | 9.05 | 8.23 |
| Arc Fault $(10^{-6})$ | **46.8** | **63.0** | **91.8** | **13.5** | **48.1** | **75.0** | **151** | **70.5** | **78.4** | **66.3** |

## 4. Experimental Results for DC Fault Arc Recognition Based on Deep Learning

*Analysis of Fault Arc Using Multi-Level Mathematical Morphological Filters*

Figure 7 illustrates the experimental design for current fault diagnosis and prediction tasks, encompassing multiple key modules. Firstly, the SRNN module utilizes a recurrent neural network (RNN) for the automatic feature extraction of current data, and captures temporal information in current data, such as the changing trend of current over time, extracting features related to faults. Additionally, the Hand-crafted Feature module serves as a data module for manually extracting features [10,13]. These manually designed features can include frequency-domain features, amplitude features, waveform features, etc., accurately expressing crucial information in the current data. In comparison to automatic feature extraction, manually extracted features are more interpretable and enhance the model's expressive capabilities and performance.

Both manually extracted features and automatically extracted features through the SRNN module, with the aim of enhancing the performance and robustness of the model. Firstly, although deep learning models, particularly recurrent neural networks (RNNs), are powerful in automatically extracting temporal data features, they often require a large amount of data to capture all relevant patterns and dependencies. In the analysis of current signals, certain important features may not be easily identified automatically, especially when there are subtle but critical patterns present in the signals. Therefore, by incorporating manual feature extraction, we ensure that these important pieces of information are not overlooked and are considered early in the model training process. Secondly, manually extracted features can serve as prior knowledge introduced into the model, helping guide the learning process, especially in scenarios with limited datasets or less obvious features. This fusion of manual and automatic feature extraction methods can accelerate the convergence speed of the model while improving its generalization ability when dealing with unseen data. Lastly, our experimental results demonstrate that the hybrid model combining manual features outperforms the model using only automatic feature extraction in multiple performance metrics. This indicates that manual feature extraction remains valuable within the current framework. Although this may introduce some dependence on the manual features, this strategy is effective in achieving higher prediction accuracy. Of course, we acknowledge that reducing reliance on manual feature extraction is a long-term goal, and we will continue exploring data-driven feature extraction methods to enhance the automation level of the model and reduce dependence on domain expertise.

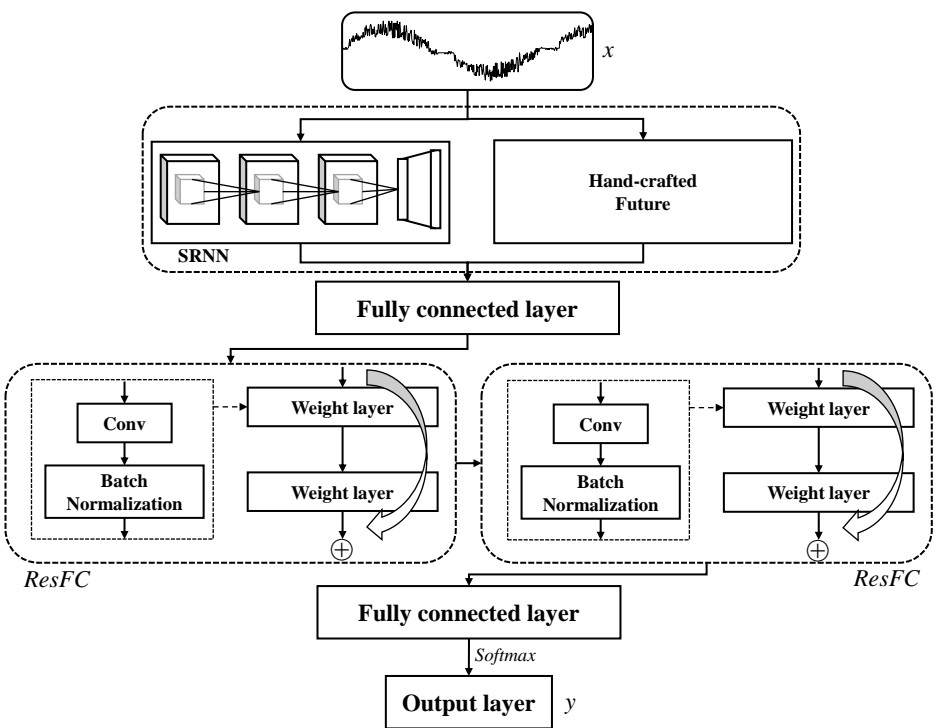

**Figure 7.** Overall architecture of the experimental design for current fault diagnosis and prediction tasks.

Furthermore, the Contact module plays a role in concatenating automatically and manually extracted features throughout the entire model. Since automatically and manually extracted features have different information expression and representation capabilities, by concatenating them, we can obtain a more comprehensive, integrated feature vector. This fusion method fully leverages the advantages of different feature extraction methods, further enhancing the model generalization capabilities and performance. In the final stage of the entire model, we employ a Fully Connected layer (FC) to map the feature vector to

the ultimate output. The FC layer, through linear and non-linear transformations, converts abstract features into concrete judgments or predictions.

Additionally, the ResFC module is shown in Figure 8, which introduces a design with residual connections into the model. Deep models often learn more complex, abstract feature representations but are susceptible to the issues of vanishing or exploding gradients. To address this problem, the ResFC module, through the introduction of skip connections, allows information to pass directly through the network without excessive interference from multiple layers. This design of residual connections effectively alleviates gradient issues, improving the model training effectiveness and convergence speed. Furthermore, to further prevent the occurrence of overfitting, the ResFC module also employs Dropout technology to randomly discard the output of some neurons.

The experimental design scheme illustrated in Figure 7 is a comprehensive framework that integrates various methods, including automatic feature extraction, manual feature extraction, feature fusion, and deep enhancement. It can extract rich and accurate features from current data, providing robust support for current fault diagnosis and prediction tasks. In the future, combining more domain knowledge and algorithmic techniques can further optimize this framework to enhance the accuracy and robustness of current fault diagnosis and prediction. For example, consider the use of multimodal fusion methods (such as joint processing of image and voice data) to improve the expressive power and discriminability of current data. Additionally, exploring more efficient and interpretable feature extraction and selection methods can achieve better results in current fault diagnosis and prediction.

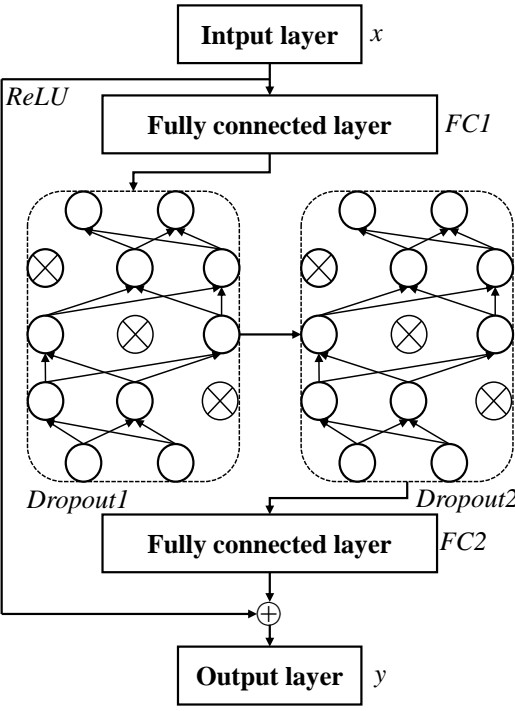

**Figure 8.** The implementation of the ResFC module includes the incorporation of Dropout for preventing overfitting.

The input to the RNN model consists of current data from 240 sampling points and manually extracted 6 feature vectors. To better express the crucial information of the current data, we extracted variance and peak-to-peak features in the time domain, as well as harmonic energy features in four frequency bands in the frequency domain. By manually extracting these feature vectors, we can accurately capture key information in the current data, laying the foundation for subsequent fault diagnosis and prediction tasks. During the iterative process, we utilize these inputs to update various hyperparameters until the network converges or reaches the maximum iteration count. By continuously adjusting

hyperparameters, we can optimize the model performance and accuracy. This process is iterative, with each iteration making the model more accurately learn the patterns and trends in the current data. After completing the design of the entire model, we label the dataset accordingly and split it into a training set and a test set in a 4:1 ratio. Dataset division is performed to assess the model generalization ability on unseen data. Then, we use the data from the training set to train the model and the data from the test set for classification and performance evaluation.

The model performance is shown in Table 5. It is noteworthy that the model performs slightly better on the test set compared to the training set. This could be attributed to the relatively smaller size of the test set. This phenomenon is caused by the limited sample size that may introduce chance variations. To provide a more detailed illustration, the ROC curve is introduced to demonstrate the model classification capability on the test set in Figure 9. It is evident that even when the false positive rate (FPR) is equal to 0, the model true positive rate (TPR) remains high. This indicates that the model maintains a high recognition accuracy while minimizing false positives. Furthermore, the area under the ROC curve approaches 1, further affirming the model's outstanding robustness and stability. The training and test results are tabulated in Table 5. The model performs slightly better on the test set than on the training set. Additionally, the ROC curve in Figure 9 confirms the excellent robustness of the model, as the TPR remains high even at the lowest false positive rate. These findings provide crucial reference points for our assessment of the model performance.

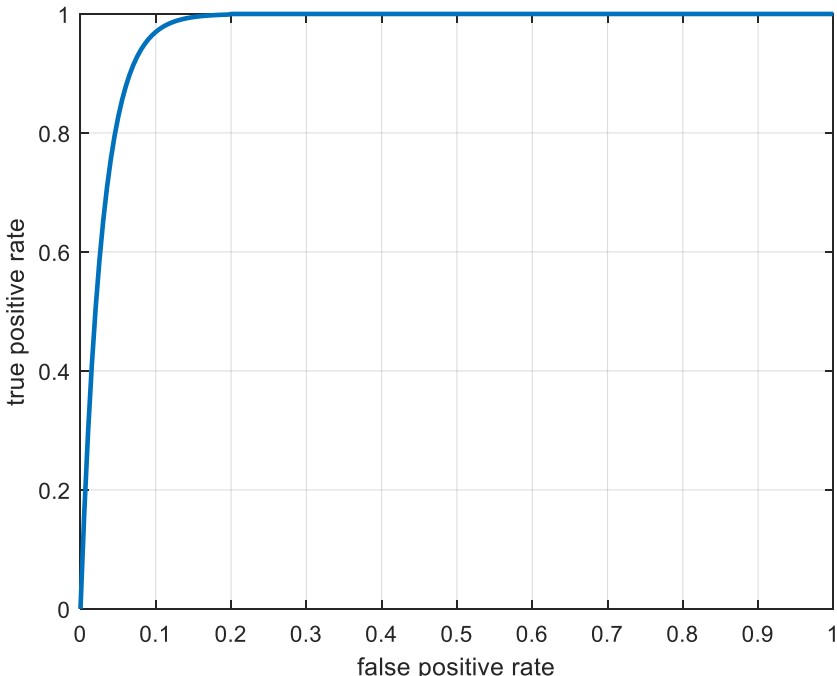

**Figure 9.** ROC curve of the model on the test set.

**Table 5.** Results of training set and test set.

|  | Loss | Accuracy |
| --- | --- | --- |
| Training set | 0.0531 | 0.9822 |
| Test set | **0.0527** | **0.9824** |

Combining the results of the above experiments, we integrated manually extracted features with those obtained through RNNs. Both sets of combined features were input into the model simultaneously for training, leading to higher accuracy on the final test set. Manual feature extraction enables us to actively select the most discriminative features

for the problem, thereby improving the model's performance. On the other hand, RNNs excel in capturing long-term dependencies in sequential data, enabling the model to better understand context and contextual information. The combination of these two features fully maximizes their respective strengths, thereby improving the overall performance and accuracy of the model.

## 5. Discussion

In our paper, we have provided detailed descriptions of the key data and the resulting values obtained from our experiments. This encompasses performance indicators of the direct current fault arc detection algorithm, including accuracy, recall rate, and the area under the ROC curve. In particular, we have emphasized achieving 98.24% accuracy on the test set and the performance of the ROC curve approaching 1, which demonstrates the efficiency and reliability of our proposed method in detecting direct current fault arcs.

To prevent arc faults, we recommend a series of guidelines for the design, installation, and operation of photovoltaic systems to minimize the occurrence of arc faults. These suggestions will include, but are not limited to:

(1). Enhancing Monitoring and Preventive Measures: We recommend regular inspection and maintenance of photovoltaic systems, with a focus on cable connections and insulation, to minimize the risk of faults.

(2). Using High-Quality Components: We recommend using high-quality and internationally standardized photovoltaic components to enhance the safety and reliability of the entire system.

(3). Installing Advanced Fault Detection Systems: We recommend integrating advanced fault detection technologies, such as our proposed detection scheme based on recurrent neural networks, into photovoltaic systems to identify and address potential arc faults promptly.

## 6. Conclusions

With the growing frequency of fire incidents caused by DC arc faults in PV systems, safety is seriously threatened. Therefore, detecting and identifying DC arc faults in PV systems holds significant practical importance. In this study, an experimental PV system established a data collection platform for DC arc faults. The mathematical morphology method was used to extract features from DC arc faults, and various morphological operators were combined to develop a classifier based on RNN. Experimental results demonstrate that this method achieves greater accuracy in classifying DC arc faults for detection purposes. Three main contributions have been made. Firstly, the mathematical morphology methods for detecting DC arcs in PV systems are adopted. Secondly, deep learning methods are employed to identify DC arcs. This approach has made significant progress in feature extraction and has achieved high accuracy. Thirdly, RNN is used for DC arc recognition. The RNN structure performs well in handling time-series tasks and achieves high-accuracy recognition by training on multiple features. The ROC curve on the test set approaches 1 from the initial state, and the accuracy on the test set remains at 98.24%, indicating the strong robustness of the proposed model. In the future, integrating other data sources, such as meteorological data and temperature sensor data, with the identification of DC arcs in photovoltaic systems will be considered to enhance the detection and prediction capabilities for abnormal situations. By exploring this prospect, valuable references for research and practice in related fields might be provided.

**Author Contributions:** Methodology, L.S. and C.L. (Chunguang Lu); software, C.L. (Chen Li); validation, Y.X. and J.Z.; data curation, L.L.; writing—original draft preparation, W.L. and X.W. All authors have read and agreed to the published version of the manuscript.

**Funding:** This work was funded in part by the Technology Project of State Grid Zhejiang Electric Power Company (Grant No. 5211YF220008), the Sanya Science and Technology Innovation Project

(Grant No. 2022KJCX47), and the Research Startup Funding from Hainan Institute of Zhejiang University (Grant No. 0210-6602-A12203).

**Institutional Review Board Statement:** Not applicable.

**Informed Consent Statement:** Not applicable.

**Data Availability Statement:** The data presented in this study are available on request from the corresponding author.

**Conflicts of Interest:** Lei Song, Chunguang Lu, Chen Li, Yongjin Xu, and Jiangming Zhang were employed by the company Marketing Service Center of State Grid Zhejiang Electric Power Co., Ltd. Lin Liu was employed by the company State Grid Hangzhou Xiaoshan District Power Supply Company. There are no conflicts of interest and the companies had no role in this study. The authors declare no conflicts of interest. The funders had no role in the design of the study; in the collection, analyses, or interpretation of data; in the writing of the manuscript; or in the decision to publish the results.

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
