# Peer review of "Arc Detection of Photovoltaic DC Faults Based on Mathematical Morphology"

_machines, doi:10.3390/machines12020134_

Round 1

Reviewer 1 Report

Comments and Suggestions for Authors

Dear authors, first of all I want to congratulate you for the work done, although I would like to suggest certain contributions that will improve your proposal.

It is important that they reinforce the abtrac with results obtained in the study.

Improve the introduction by clearly highlighting the objective and novelty of the study.

There is a more important part that focuses on the concept of arc failure itself.

I think it is necessary to clarify the causes of the failure by giving values of the operating parameters of the photovoltaic generator that are distorted.

Another important aspect and since it may be due to the malfunction of the installation, connect with the operating standard for photovoltaic systems, 61724, and say which sensors should be installed and what operations should be carried out with the monitored values.

In this sense, it would be important for the authors to include in the review of the state of the art studies and proposals to include in the standard.

It will also be necessary for the conclusions to incorporate some values of the results obtained and recommendations to avoid arc failures.

Author Response

Thank you for your careful review. We have made revisions based on your comments. Please review the revised draft and response accordingly.

Reviewer 2 Report

Comments and Suggestions for Authors

Some small errors are detected:

Rows 35-36 It's repeated.

Rows 47-48 It's repeated.

Rows 63 …microgirds. a multichracteristics… must be changed by ……microgirds. A multichracteristics…

Row 141: Introduce “GASF”, SE-DCGAN” and “SAF” terms, buy they are not defined yet.

Some discussion

Row 313. Ther are manual procedures by to extract some important features from the signal. And in 315 rows the importance of them with respect to automatic extraction techniques are expressed. A big limitation to the usability of this technique consists of the dependence from manual extraction of information.

Additionally, I do not found some way to do some experiments in order to do reproducible test. Besides, one experiment it's not enough to do some important conclusions. I encourage doing more test and advance in order to avoid do a manual extraction of information, searching to have an automatic process to implement into a PV system.

Author Response

(The authors gave the same response as above.)

Round 2

Reviewer 1 Report

Comments and Suggestions for Authors

I thank the authors for all the work done trying to respond to my suggestions.

I believe that the work has now been completed more, although to make it more useful, would you be so kind as to define what measures or values are necessary to maintain the useful life of the systems to anticipate arc failure?

Author Response

Thank you for your careful review and valuable feedback. These are of great help in improving this manuscript. We have made the following modifications to the manuscript:

-1-. The abstract has been reorganized.

-2-. Improved the logical structure of the main text, clarified the mathematical relationships between different variables, and provided physical explanations for some variables.

-3-. Removed unnecessary references and added relevant research citations to this work.

-4-. Continuously corrected some spelling errors.

We hope you will recognize these improvements, and we appreciate your re-evaluation.

Reviewer 2 Report

Comments and Suggestions for Authors

Thanks very much by improve the paper.

Author Response

(The authors gave the same response as above.)
